# Identification of Novel Src Inhibitors: Pharmacophore-Based Virtual Screening, Molecular Docking and Molecular Dynamics Simulations

**DOI:** 10.3390/molecules25184094

**Published:** 2020-09-08

**Authors:** Yi Zhang, Ting-jian Zhang, Shun Tu, Zhen-hao Zhang, Fan-hao Meng

**Affiliations:** School of Pharmacy, China Medical University, Shenyang 110122, China; cruckzhang0304@163.com (Y.Z.); todayfy@outlook.com (T.-j.Z.); tushun2018@163.com (S.T.); zhangzhenhao314@163.com (Z.-h.Z.)

**Keywords:** Src inhibitors, pharmacophore model, virtual screening, molecular docking, molecular dynamics simulations

## Abstract

Src plays a crucial role in many signaling pathways and contributes to a variety of cancers. Therefore, Src has long been considered an attractive drug target in oncology. However, the development of Src inhibitors with selectivity and novelty has been challenging. In the present study, pharmacophore-based virtual screening and molecular docking were carried out to identify potential Src inhibitors. A total of 891 molecules were obtained after pharmacophore-based virtual screening, and 10 molecules with high docking scores and strong interactions were selected as potential active molecules for further study. Absorption, distribution, metabolism, elimination and toxicity (ADMET) property evaluation was used to ascertain the drug-like properties of the obtained molecules. The proposed inhibitor–protein complexes were further subjected to molecular dynamics (MD) simulations involving root-mean-square deviation and root-mean-square fluctuation to explore the binding mode stability inside active pockets. Finally, two molecules (ZINC3214460 and ZINC1380384) were obtained as potential lead compounds against Src kinase. All these analyses provide a reference for the further development of novel Src inhibitors.

## 1. Introduction

The Src family kinases (SFKs) are a family of non-receptor tyrosine kinases, which are involved in a wide variety of essential functions to sustain cellular homeostasis, where they regulate cell cycle progression, motility, proliferation, differentiation and survival, among other cellular processes [1]. As a prototypical member of the SFKs, Src contains Yes, Fyn, Lyn, Lck, Hck, Fgr, Yrk, Frk and Blk kinases [2]. Src consists of four homology domains (SH1, SH2, SH3 and SH4) and a unique domain (Figure 1). The SH1 domain (also called the catalytic domain) is composed of two subdomains (generally termed N-terminal and C-terminal lobes) separated by a cleft. The N-terminal lobe contains the highly conserved hinge region that is implicated in the interaction with the ATP-adenine ring and to which most of the Src inhibitors anchor through hydrogen bonding. The C-terminal lobe is larger, comprises an activation loop that contains a tyrosine residue that can be autophosphorylated (Tyr419 in human c-Src) and is the positive regulatory site responsible for maximizing kinase activity. The phosphorylation of this residue stabilizes the kinases in an active conformation accessible to ATP and substrates. On the contrary, when another tyrosine residue located in the C-terminal lobe tail (Tyr530 in human c-Src) is phosphorylated, a closed conformation is induced [3]. The SH2 and SH3 domains regulate the Src catalytic activity through both intramolecular and protein–protein interactions. The SH4 domain is a 15-amino acid sequence whose myristoylation allows the binding of Src members to the inner surface of the plasma membrane. The unique domain is included in the N-terminal segment of the proteins, together with SH4, and is composed of 50–70 residues. Unlike the SH domains, it displays the greatest divergence among the SFKs and thus probably contributes to the differentiation of their biological functions [4]. Src is a central signaling hub that can be activated by many factors, including immune-response receptors, integrins and other adhesion receptors, receptor protein tyrosine kinases, G protein-coupled receptors and cytokine receptors [5]. In normal cells, Src is only transiently activated during the multiple cellular events in which it is involved. Conversely, Src is overexpressed and/or hyperactivated in a large variety of solid tumors and is probably a strong promoting factor for the development of metastatic cancer phenotypes [6]. Src is responsible for many human cancers such as lung [7], neuronal [8], ovarian [9], esophageal [10] and gastric cancers [11], as well as melanoma [12] and Kaposi’s sarcoma [13]. Due to its involvement in many cellular processes related to cancer development, Src has long been considered a potential drug target in oncology.

The Src inhibitors developed to date are generally categorized into three major classes: (1) tyrosine kinase activity inhibitors (ATP-competitive inhibitors); (2) protein–protein interaction inhibitors (SH2, SH3 or substrate-binding domain); (3) enzyme destabilizers that provide a correlation between Src and its united molecular chaperone, i.e., heat shock protein 90 (Hsp90) [14,15]. The search for small molecules with an inhibitory activity toward Src kinases constitutes a growing field of study. Several compounds have entered clinical trials, with two compounds ultimately approved by the FDA: dasatinib, approved in 2006, and bosutinib, approved in 2012 [16]. However, dasatinib is known to inhibit over 40 kinases, while bosutinib inhibits over 45 kinases, making it impossible to use these compounds as selective mechanistic probes for Src-dependent pharmacology [17,18]. Furthermore, most Src inhibitors reported share similar scaffolds such as pyrazolo [3,4-d] pyrimidine, quinoline and quinazoline (Figure 2). To this end, it is meaningful to find more effective and selective Src inhibitors with new chemical scaffolds.

In this work, we report an integrated screening method containing pharmacophore-based virtual screening; molecular docking; absorption, distribution, metabolism, elimination and toxicity (ADMET) prediction; and molecular dynamics (MD) simulations to find novel Src inhibitors.

## 2. Results and Discussion

### 2.1. Preparation of Chemical Database

Prior to performing the virtual screening, the database needed to undergo several filtering and preparation steps to reduce the enormous number of compounds [19]. All the selected ligands were downloaded from the ZINC database (https://zinc.docking.org/) using filters such as “in-stock” and “drug-like”, and the selection of ligands was performed based on Lipinski’s rule of five (molecular weight limit of 300 to 600 Da, hydrogen-bond acceptor limit of 10, hydrogen-bond donor limit of 5, rotatable-bond limit of 7, and log P limit of 5). To produce a more refined and precise set of chemical data, some built-in functions such as the “partial charges” and “energy minimize” tools of the Molecular Operating Environment software (MOE, Version 2015.10) were applied on the data set. The resulting database comprised 1,033,419 molecules with lowest energy in 3D format.

### 2.2. Generation and Validation of Pharmacophore Model

The protein–ligand complex serves as the starting template for this modeling, wherein intermolecular interactions are perceived as feature points for subsequent virtual screening. Most often, a single protein–ligand complex is used as the template to align and score the database molecules, from which the best-fitted molecules are prioritized as potential hits [20]. Based on the crystal structure of Src kinase selected (PDB ID: 3F3V), five key pharmacophore features were generated, including one hydrogen-bond donor (Don), two hydrogen-bond acceptors (Acc), one hydrophobic and aromatic center (Hyd/Aro) and one aromatic center (Aro). As shown in Figure 3, the pharmacophore model was designed in consideration of the binding poses of the original ligand (RL45, yellow sticks). Three hydrogen-bond features were present for the ligand–protein complex: the hydrogen-bond donor (F1, purple sphere) of RL45 interacted with Glu310; the hydrogen-bond acceptors (F2 and F3, cyan sphere) of RL45 interacted with Asp404 and Met341, respectively. One hydrophobic and aromatic center (F4, yellow sphere) and one aromatic center (F5, orange sphere) were also present for the ligand.

For the validation of generated pharmacophore model, a test database was built including 18 Src reported inhibitors and 18 collected decoy molecules, which can be seen in the Appendix A. The test database was then subjected to screening against the pharmacophore model to validate its precision. As a result, we obtained 14 active molecules as hits, and none of the inactive molecules was mapped to the pharmacophore model. The results from the test database revealed the precision of the generated pharmacophore model.

### 2.3. Pharmacophore-Based Virtual Screening

The pharmacophore-model-based screening of databases has been considered as an important tool for computer-aided drug discovery techniques and provides information about geometric and electronic features that are involved in interaction with receptors [21]. In this part, the chemical database comprised 1,033,419 molecules with lowest energy in 3D format and was generated by applying various filters. We utilized the protein–ligand complex reported for Src for pharmacophore model generation and performed virtual screening of the prepared database to find out the best matches against the model. As a result, 891 molecules were obtained as hits on the basis of pharmacophore features.

### 2.4. Molecular Docking

The hits obtained from the pharmacophore-based virtual screening were subjected to molecular docking studies; the top 10 molecules with the highest docking scores were selected for the study of binding modes (Table 1). Among these 10 molecules, ZINC23247639 and ZINC10479320 have been reported as broad-spectrum kinase inhibitors that interacted with the highly conserved ATP-binding sites of many human protein kinases [22,23,24]; thus, we filtered them out in this investigation. As we know, hydrogen bonds established between receptor and ligand play a major role in the functionality and stability of the complex. Hence, we observed the dominant hydrogen-bond interactions between the groups of the other eight hit molecules and the residues of the active site, and then, ZINC3214460 and ZINC1380384 caught our attention.

Using the default GBVI/WSA dG as a docking function in the MOE software, ZINC3214460 and ZINC1380384′s docking scores were calculated as −9.6287 and −8.9096 kcal/mol, respectively. The binding interactions of ZINC3214460 and ZINC1380384 were illustrated by the PyMOL [25] software (Figure 4). As we can see, some key amino acid residues were involved in hydrogen-bonding interactions with ZINC3214460; a carbonyl group accepted an H-bond from the Asp404, and an N atom of the isoxazole formed an H-bond with Met341. The H-bond distances were 2.87 and 3.19 Å, respectively, and hydrogen-bonding energy components contributed −4.6 kcal/mol to the binding. Correspondingly, ZINC1380384 formed three H-bond interactions with the kinase. The amide fragment formed two H-bonds with Asp404 and Glu310, respectively, and an N atom of the benzimidazole accepted an H-bond from a Met341 residue. The hydrogen-bonding energy components contributed −6.9 kcal/mol to the binding. On the basis of good binding energies and their pattern of binding interaction with active pocket, two selected molecules showed strong interaction with the key amino acid residues of Src kinase. Furthermore, these two molecules have not been reported as kinase inhibitors or related with cancer yet, so our attention is focused on ZINC3214460 and ZINC1380384, which were further subjected to ADMET prediction and molecular dynamics simulations.

### 2.5. ADMET Prediction

The lead compounds in drug development were found to have favorable absorption, distribution, metabolism, elimination and toxicity (ADMET) properties [26]. Pharmacokinetic properties predict the drug-likeness of ligand molecules. Therefore, the ADMET properties of molecules are essential for the development of an effective druggable molecule. In this section, ADMET characteristics such as buffer solubility, blood–brain barrier penetration (BBB), Caco-2 permeability, human intestinal absorption (HIA), plasma protein binding (PPB), cytochrome P450 2D6 (CYP2D6) modulation and hERG inhibition were studied for ZINC3214460, ZINC1380384, dasatinib and bosutinib. The results are summarized in Table 2. Solubility and human intestinal absorption are two key factors that affect oral bioavailability. We found that ZINC3214460 and ZINC1380384 show extremely high values of solubility. Low blood–brain barrier permeability was found, which served in reducing the side effects and toxicity to the brain, and the values of all the compounds are less than 1 (C.brain/C.blood < 1), suggesting that they are inactive in the CNS (central nervous system). Caco-2 permeability was used to evaluate the suitability of compounds for oral dosing, and the proposed compounds have slightly worse human intestinal permeability than two drugs approved by the FDA. In addition, the comparable intestinal absorption (HIA) for dasatinib and bosutinib indicated that two proposed compounds possess good bioavailability. The high plasma-protein binding of ZINC3214460 means a long half-life and stable efficacy, which could maintain a durable potency and adequate stability of the compound. The inhibition of CYP2D6 by a drug constitutes the majority of cases of drug–drug interaction. It was found that none of the compounds may inhibit CYP2D6. The cardiotoxicity may be related with the high risk of hERG inhibition, as shown in Table 2; ZINC3214460 has a low risk of inhibiting hERG, meaning that it has little chance of causing cardiac problems. However, according to the high-risk level of hERG inhibition, the potential cardiotoxicity of ZINC1380384 should be considered in the future. More ADMET prediction data for these two proposed molecules can be found in the Appendix A.

### 2.6. Molecular Dynamics Simulations

MD simulations were conducted to check the stability of the complexes predicted by molecular docking. After 50 ns MD simulations, the root-mean-square deviation (RMSD) of the backbone of Src kinase and the ligands at 300 K was plotted against time (ns). As can be seen in Figure 5, the RMSDs of 3F3V-ZINC3214460 and 3F3V-ZINC1380384 were discovered to be relatively stable at about 0.27 and 0.22 nm, respectively. There were some fluctuations in the beginning, and then, the complexes gradually tended to equilibrium until the time reached 25 ns of simulation. The RMSD values of RL45 and ZINC1380384 in the binding site of Src are similar; however, the 3F3V-ZINC3214460 complex has a larger RMSD value at around 0.30 nm. The root-mean-square fluctuation (RMSF) is an important parameter that yields data about the structural adaptability of every residue in the system. The RMSF values for all the residues in the 3F3V-ZINC3214460 and 3F3V-ZINC1380384 complexes were calculated (Figure 6). In general, the Src inhibitors are settled to the highly conserved region of the protein (Glu339, Asp348 and Asp404 in the affinity pocket and Glu310 and Leu317 in the hinge region) through hydrogen bonds, van der Waals forces and hydrophobic interaction, etc. We found that the RMSF fluctuation values of the conserved amino acids (e.g., Glu310, Leu317 and Glu339) in 3F3V-ZINC3214460 and 3F3V-ZINC1380384 were lower than those in 3F3V-RL45, reflecting that the two proposed molecules could form stronger interactions with these conserved residues. The proteins showed distinct behavior with a varied magnitude of fluctuations at the loop region (e.g., Ser522 and Tyr527), which could be used to discover more selective inhibitors of Src. Based on the above analysis, we can conclude that these two selected molecules matched very well with the Src binding pocket, suggesting reasonable binding modes.

## 3. Materials and Methods

### 3.1. Generation and Validation of Pharmacophore Model

In this study, a protein–ligand complex-based pharmacophore model was generated by using the pharmacophore query editor protocol of MOE. Binding interactions induce significant chemical features, which were taken into account for the creation of the pharmacophore model [27]. To generate a pharmacophore model with good quality, MOE utilizes an in-built set of pharmacophore features including hydrogen-bond donor (Don), hydrogen-bond acceptor (Acc), aromatic center (Aro), Pi ring center (PiR), aromatic ring or Pi ring normal (PiN), hydrophobic atom (HydA), anionic atom (Ani), cationic atom (Cat) and so on. We summarized all the binding interactions of Src complexes in the DFG-out state from the RCSB Protein Data Bank (PDB, https://www.rcsb.org/), and several common features (e.g., hydrophobic interactions, hydrogen bonding modes and catalytic residues) were utilized for generating the model. Finally, the crystal structure of Src kinase in complex with a substrate-based inhibitor, 1-(4-((6-aminoquinazolin-4-yl) amino) phenyl)-3-(3-(tert-butyl)-1-(m-tolyl)-1H-pyrazol-5-yl) urea (RL45) (PDB ID: 3F3V; resolution, 2.6 Å), was chosen for the creation of a complex-based pharmacophore system. The inhibitor RL45 had strong interaction with key amino acid residues of Src kinase (i.e., Asp404, Glu310 and Met341) and was further used as a reference in molecular docking (Figure 7). The binding interactions of RL45 with the active pocket of Src are illustrated by the PyMOL software (2.3.2).

The generated pharmacophore model was validated via a test database including 18 Src reported inhibitors and 18 collected decoy molecules. The molecules of Src inhibitors were downloaded from DrugBank (https://www.drugbank.ca/). The decoy molecules refer to those compounds with reported activities not related to Src, whereas their physical properties, including molecular weight, number of hydrogen-bond donors and acceptor, number of rotatable bonds, and Log P were similar to those of known Src inhibitors. The test database can be seen in the Appendix A.

### 3.2. Pharmacophore-Based Virtual Screening

In Computer-Aided Drug Designing (CADD), virtual screening is one of the time-saving methods for the discovery of novel, potent and drug-like compounds [28]. Pharmacophore models have the advantage that they can be used not only to identify novel active compounds in virtual screening but also for anti-target modeling to avoid side-effects resulting from off-target activity [29]. Especially, when structural information about the target protein or the ligand’s active conformation is available, pharmacophore-based models are superior to docking and quantitative structure–activity relationship (QSAR) methods [30]. Based on the pharmacophore model generated above, virtual screening was conducted by a pharmacophore search protocol in MOE with an EHT scheme. MOE’s pharmacophore search is used to apply a query to a database of molecular conformations and report those conformations as hits that satisfy the pharmacophore features. The hit molecules were preferred and kept in a separate database for the further evaluation of interactions.

### 3.3. Molecular Docking

The crystal structure of Src kinase was obtained from the RCSB Protein Data Bank (PDB ID: 3F3V; resolution, 2.6 Å). The protein was prepared for docking using the quickprep tool of MOE, including correcting structural issues, protonating the structure, deleting unbound water molecules and minimizing the structure to a specified gradient, to make the pocket available for the docking of new molecules. The original ligand (RL45) was used to define the binding site of the Src active pocket. For the docking parameters, we set the force field to MMFF94x and used the triangle matcher placement algorithm [31], which returned thirty poses; we also used the Rigid Receptor refinement method, which returned five poses. The London dG method was applied to score the poses in both steps [32]. By studying the top-scored docking poses, only the molecular poses for which the binding modes satisfied the pharmacophore features were retained. Each molecule with the highest docking score was regarded as a docking result for further analysis.

### 3.4. ADMET Prediction

Owing to poor pharmacokinetic parameters, many drugs have not passed through clinical trial stages [33]. Thus, it is necessary to predict the pharmacokinetics and toxicity of newly obtained molecules, which were selected from the docking results for further analysis. In this section, the Pre-ADMET server application (https://preadmet.bmdrc.kr) was used. The Pre-ADMET approach is based on different classes of molecular parameters that are considered for generating quantitative structure properties [34].

### 3.5. Molecular Dynamics Simulations

Based on the docking results, the best-posed complex was subjected to MD simulation studies using the Groningen Machine for Chemicals Simulations (GROMACS) 5.0 package [35] with a CHARMM36 force field [36] under periodic boundary conditions for molecules. Ligand topology files were generated using the CHARMM General Force Field [37]. The charge of the system was neutralized by the addition of the ions. The energy was minimized using a steepest-gradient method to remove any close contacts. The particle mesh Ewald (PME) method was employed for energy calculation and for electrostatic and Van der Waals interactions. The systems were equilibrated in the NVT ensemble for 50,000 steps, followed by equilibration in the NPT ensemble for an additional 50,000 steps. Finally, 50 ns molecular dynamics simulations were performed at 300 K with a 2.0 fs time step, and coordinates were saved every picosecond for analysis [38,39].

## 4. Conclusions

In this study, in order to find novel Src inhibitors, an integrated screening method was employed; through pharmacophore-based virtual screening and molecular docking, the top 10 molecules with good binding scores were selected for the study of binding modes. ADMET prediction and molecular dynamics simulations were used to predict the pharmacokinetic properties and stabilities of proposed ligand–protein complexes. Finally, two molecules (ZINC3214460 and ZINC1380384) were selected with excellent properties and stable binding modes. In addition, ZINC1380384, possessing a novel benzo [d] imidazole scaffold, was valuable for further optimization and provides a reference for the development of novel potent Src inhibitors.

## Figures and Tables

**Figure 1 molecules-25-04094-f001:**
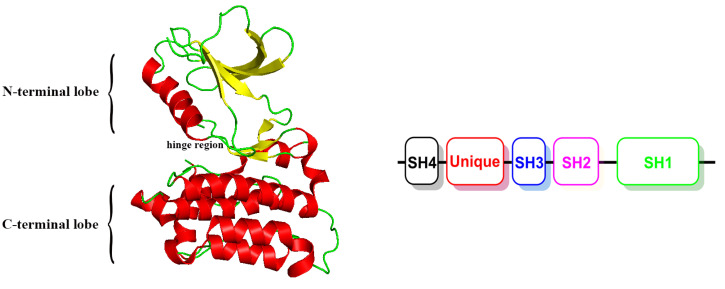
The crystal structure of the Src kinase and schematic domain structure.

**Figure 2 molecules-25-04094-f002:**
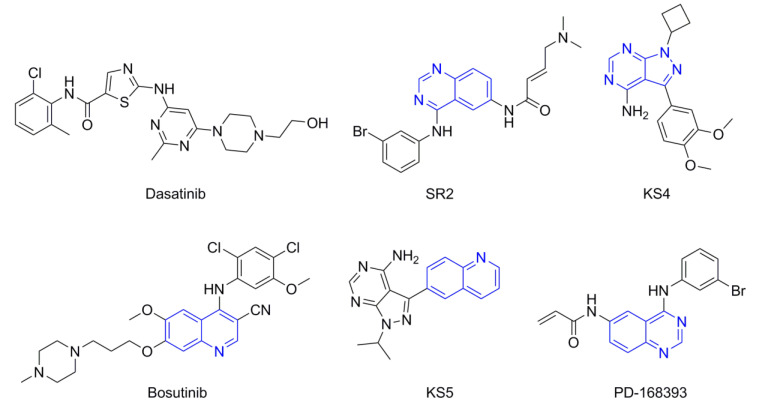
Chemical structures of previously reported Src inhibitors.

**Figure 3 molecules-25-04094-f003:**
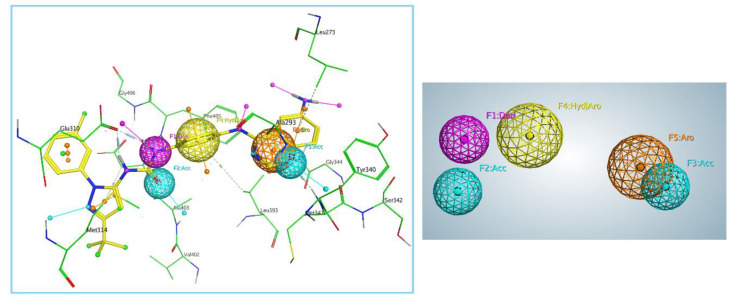
Pharmacophore features generated by Molecular Operating Environment (MOE).

**Figure 4 molecules-25-04094-f004:**
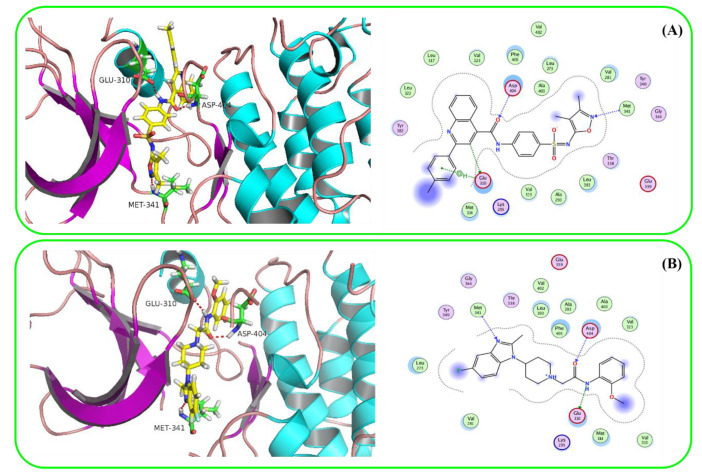
Binding interactions of two hit molecules with active pocket of Src kinase. The hit molecules ZINC3214460 (**A**) and ZINC1380384 (**B**) are displayed in yellow sticks, and catalytic residues are displayed in green sticks. Hydrogen bonds are shown as red dashes.

**Figure 5 molecules-25-04094-f005:**
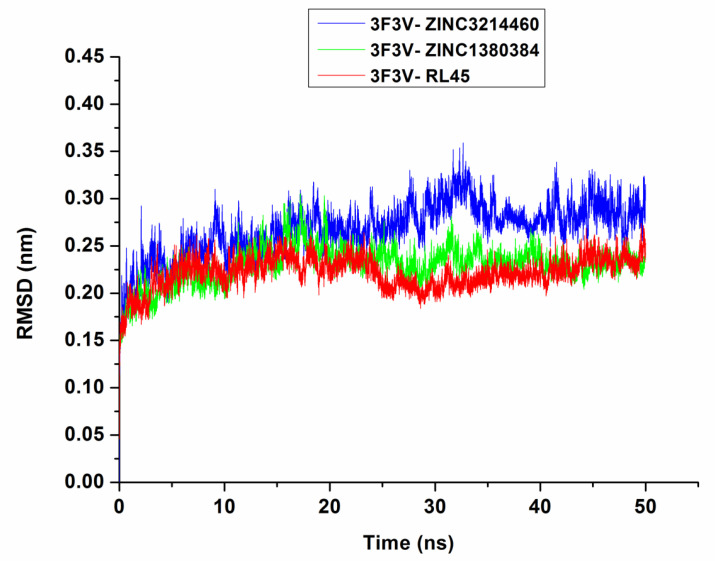
The root-mean-square deviation (RMSD) trajectories of 3F3V–inhibitor complexes during 50 ns simulations.

**Figure 6 molecules-25-04094-f006:**
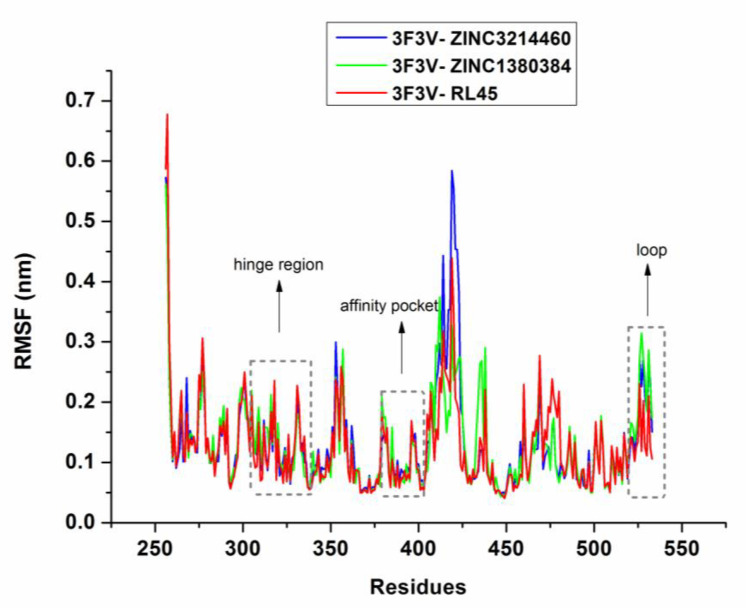
The root-mean-square fluctuation (RMSF) maps of 3F3V–inhibitor complexes during simulations.

**Figure 7 molecules-25-04094-f007:**
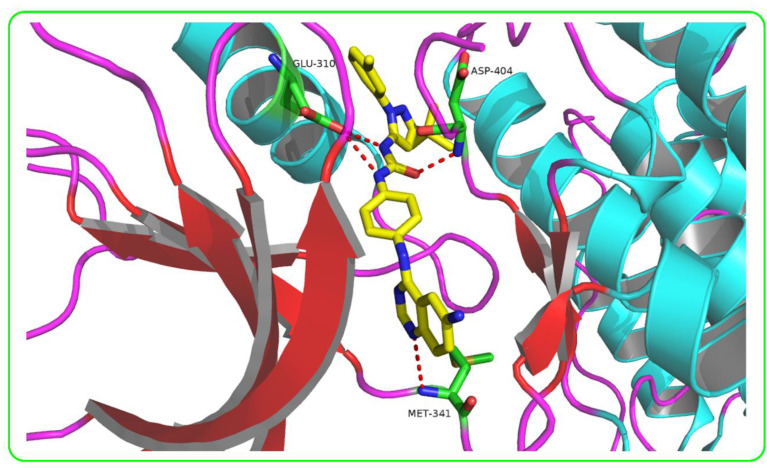
Binding interaction of RL45 with active pocket of Src kinase. The ligand RL45 is displayed in yellow sticks, and catalytic residues are displayed in green sticks. Hydrogen bonds are shown as red dashes.

**Table 1 molecules-25-04094-t001:** The structures and docking results of molecules.

ZINC ID	Structure	Src Docking Score (kcal/mol)
ZINC3214460	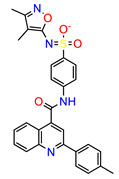	−9.6287
ZINC61925676	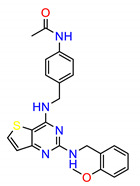	−9.1879
ZINC58158745	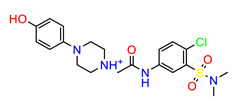	−9.1320
ZINC12075400	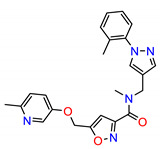	−8.9992
ZINC1380384	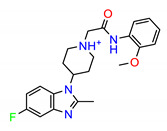	−8.9096
ZINC12853028	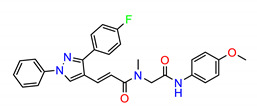	−8.7889
ZINC23247639	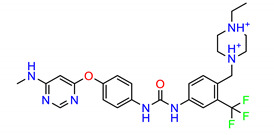	−8.7219
ZINC949873	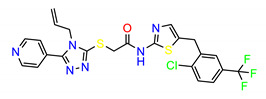	−8.5887
ZINC36389462	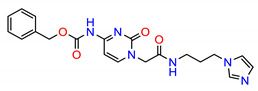	−8.5816
ZINC10479320	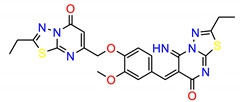	−8.5090

**Table 2 molecules-25-04094-t002:** The absorption, distribution, metabolism, elimination and toxicity (ADMET) prediction for the investigated compounds.

Compound	Buffer Solubility ^1^	BBB ^2^	Caco-2 ^3^	HIA ^4^	PPB ^5^	CYP2D6 Inhibition	hERG Inhibition
**ZINC3214460**	81.69	0.01036	18.87	97.41	100	None	Low risk
**ZINC1380384**	3735.39	0.4491	26.62	92.11	34.90	None	High risk
**Dasatinib**	0.3113	0.03504	32.01	93.59	70.29	None	Medium risk
**Bosutinib**	5.500	0.06055	50.35	97.23	85.07	None	Medium risk

^1^ Buffer solubility: water solubility in buffer system (SK atomic types, mg/L), ^2^ BBB: blood–brain barrier penetration (C.brain/C.blood), ^3^ Caco-2: in vitro Caco-2 cell permeability (nm/sec), ^4^ HIA: human intestinal absorption (%), ^5^ PPB: plasma protein binding (%).

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
