# Peer review of "Identification of Novel Src Inhibitors: Pharmacophore-Based Virtual Screening, Molecular Docking and Molecular Dynamics Simulations"

_molecules, 2020, doi:10.3390/molecules25184094_

Round 1

Reviewer 1 Report

The authors report an interesting computational study for the identification of prospective Src inhibitors, which could be suitable compounds for further investigation in cancer drug discovery. A pharmacophore model was built based on the binding mode of a known inhibitor and, then, a virtual screening was performed to identify promising hits. Two promising virtual hits were, then, subjected to molecular dynamics simulations. The two most promising compounds displayed good ADMET properties and relative stability in the MD simulations compared to marketed kinase inhibitors and the crystallographic Src inhibitor. Therefore, these are worthwhile compounds for further experimental investigation. 

The manuscript can be improved as detailed below:

Abstract: "Finally, two 21 molecules (ZINC3214460 and ZINC1380384) were obtained as novel potent Src inhibitors." This sentence is very problematic. The work does not report enzyme inhibition studies, so, no inhibitors were obtained. In silico studies alone do allow the authors to state that novel and potent inhibitors were obtained. This could be valid only if enzyme inhibition assays were performed. 

This sentence must be clarified "The high plasma protein binding of ZINC3214460 means long half-life and stable efficacy, which are accordance with the irreversible binding since the compound need to possess adequate intrinsic stability to maintain durable potency." Is the molecule a irreversible inhibitor of Src? The sentence is confusing.

How the ZINC database was filtered to reach the final database for the virtual screening? Did the authors focus in any ZINC subset? This should be detailed in the manuscript.

Author Response

Point 1: Abstract: "Finally, two 21 molecules (ZINC3214460 and ZINC1380384) were obtained as novel potent Src inhibitors." This sentence is very problematic. The work does not report enzyme inhibition studies, so, no inhibitors were obtained. In silico studies alone do allow the authors to state that novel and potent inhibitors were obtained. This could be valid only if enzyme inhibition assays were performed.

 Response 1: It is really true as Reviewer suggested that the sentence is very problematic, and we have made correction: "Finally, two molecules (ZINC3214460 and ZINC1380384) were obtained as potential lead compounds against Src kinase." Line 21-22.

Point 2: This sentence must be clarified "The high plasma protein binding of ZINC3214460 means long half-life and stable efficacy, which are accordance with the irreversible binding since the compound need to possess adequate intrinsic stability to maintain durable potency." Is the molecule a irreversible inhibitor of Src? The sentence is confusing.

Response 2: As Reviewer suggested, the statements were corrected as: "The high plasma protein binding of ZINC3214460 means long half-life and stable efficacy, which could maintain durable potency and adequate stability of the compound." Line 168-170.

Point 3: How the ZINC database was filtered to reach the final database for the virtual screening? Did the authors focus in any ZINC subset? This should be detailed in the manuscript.

Response 3: We have made correction according to the Reviewer’s comments. The ZINC database was generated from a drug-like subset of purchasable compounds as described in Line 82-90. "Prior to performing the virtual screening, the database needed to undergo several filtering and preparation steps to reduce the enormous number of compounds [19]. All the selected ligands were downloaded from ZINC database (https://zinc.docking.org/) using filters such as “in-stock” and “drug-like”, and the selection of ligands was done based on Lipinski’s rule of five (molecular weight limit of 300 to 600 Da, hydrogen bond acceptor limit of 10, hydrogen bond donor limit of 5, rotatable bond limit of 7, log P limit of 5). To make more refined and precise set of chemical database, some built-in functions like partial charges and energy minimize tools of Molecular Operating Environment software (MOE, Version 2015.10) were applied on data set. The resulting database comprises of 1,033,419 molecules with lowest energy in 3D format."

Special thanks to you for your good comments.

Reference

[19] Gentile, F.; Barakat, K. H.; Tuszynski, J. A., Computational Characterization of Small Molecules Binding to the Human XPF Active Site and Virtual Screening to Identify Potential New DNA Repair Inhibitors Targeting the ERCC1-XPF Endonuclease. International journal of molecular sciences 2018, 19, 1328-1340, doi: 10.3390/ijms19051328.

Reviewer 2 Report

In this manuscript, pharmacophore-based virtual screening and molecular docking were carried out to identify potential Src inhibitors. The proposed inhibitor-protein complexes were further subjected to molecular dynamics (MD) simulations and two molecules were obtained as novel potent Src inhibitors. The methods and results are sound. Several minor points should be addressed. 

1. The authors predicted the Caco-2 permeability, p450, and hERG inhibition. Why were these parameters chosen for this study? The authors may like to explain it in more detail. 

2. The authors claimed that computer-aided drug design has attracted attention in drug discovery and suggested that pharmacophore-based VS is a good choice to discover Src inhibitors. It is recommended to describe in more detail about the merits of pharmacophore-based VS and illustrate why pharmacophore-based VS was chosen for this study instead of other methods. The following references might be useful. 

1.Drug Discovery Today: Technologies 10.3 (2013): e395-e401. 
2. Journal of Chemical Information and Modeling 45.1 (2005): 160-169. 
3. European Journal of Medicinal Chemistry, 2018, 143: 1021-1027. 
4.Methods 2015, 71, 1. 
5. Chemical Science 2.9 (2011): 1656-1665. 

3. Some formatting errors need to be revised, such as line 66, the reference (3, 4-d) 

Overall, I recommend publication of this manuscript after minor revision. 

Author Response

Point 1: The authors predicted the Caco-2 permeability, p450, and hERG inhibition. Why were these parameters chosen for this study? The authors may like to explain it in more detail. ʉ۬

Response 1: As Reviewer suggested, the reasons of pharmacokinetic parameters chosen were added as: "The lead compounds in drug development were found to have favorable absorption, distribution, metabolism, elimination and toxicity (ADMET) properties [26]. Pharmacokinetic properties predict the drug-likeness of ligand molecules. Therefore, ADMET properties of molecules are essential for development of an effective druggable molecule." Line 152-155.

Point 2: The authors claimed that computer-aided drug design has attracted attention in drug discovery and suggested that pharmacophore-based VS is a good choice to discover Src inhibitors. It is recommended to describe in more detail about the merits of pharmacophore-based VS and illustrate why pharmacophore-based VS was chosen for this study instead of other methods.

Response 2: As Reviewer suggested, the detail about the merits of pharmacophore-based VS are illustrated as: "Pharmacophore models have the advantage that they can not only be used to identify novel active compounds in virtual screening, but also for anti-target modeling to avoid side-effects resulting from off-target activity [29]. Especially, when structural information about the target protein or the ligand’s active conformation is available, pharmacophore-based models are superior to docking and quantitative structure activity relationship (QSAR) methods [30]." Line 237-242.

Point 3: Some formatting errors need to be revised, such as line 66, the reference (3, 4-d)

Response 3: "pyrazolo [3, 4-d] pyrimidine" is a chemical name of the chemical structure moiety, it was not the reference actually. Considering the Reviewer’s suggestion, we have revised the format of chemical name as "pyrazolo [3,4-d] pyrimidine", Line 67.

Special thanks to you for your good comments.

Reference

[26] Clark, R. D., Predicting mammalian metabolism and toxicity of pesticides in silico. Pest management science 2018, 74, 1992-2003, doi: 10.1002/ps.4935.

[29] Schuster, D., 3D pharmacophores as tools for activity profiling. Drug discovery today. Technologies 2010, 7, e205-211, doi: 10.1016/j.ddtec.2010.11.006.

[30] Drwal, M. N.; Griffith, R., Combination of ligand- and structure-based methods in virtual screening. Drug discovery today. Technologies 2013, 10, e395-401, doi: 10.1016/j.ddtec.2013.02.002.